# Hypothalamic–Pituitary and Adipose Tissue Responses to the Effect of Resistin in Sheep: The Integration of Leptin and Resistin Signaling Involving a Suppressor of Cytokine Signaling 3 and the Long Form of the Leptin Receptor

**DOI:** 10.3390/nu11092180

**Published:** 2019-09-11

**Authors:** Dorota Anna Zieba, Weronika Biernat, Malgorzata Szczesna, Katarzyna Kirsz, Tomasz Misztal

**Affiliations:** 1Department of Animal Biotechnology, Laboratory of Biotechnology and Genomics, University of Agriculture in Krakow, 30-248 Krakow, Poland; 2Department of Animal Physiology, The Kielanowski Institute of Animal Physiology and Nutrition, Polish Academy of Sciences, Instytucka 3, 05-110 Jabłonna, Poland

**Keywords:** leptin, leptin receptor, leptin resistance, photoperiod, resistin, sheep, SOCS-3

## Abstract

We hypothesized that resistin is engaged in the development of leptin central insensitivity/resistance in sheep, which is a unique animal model to explore reversible leptin resistance. Thirty Polish Longwool ewes, which were ovariectomized with estrogen replacement, were used. Treatments consisted of the intravenous injection of control (saline) or recombinant bovine resistin (rbresistin): control (Control; *n* = 10), a low dose of rbresistin (R1; 1.0 μg/kg body weight (BW); *n* = 10), and a high dose of rbresistin (R2; 10.0 μg/kg BW; *n* = 10). The studies were performed during short-day (SD) and long-day (LD) photoperiods. Leptin and resistin concentrations were determined. Expression levels of a suppressor of cytokine signaling (SOCS)-3 and the long form of the leptin receptor (LeptRb) were determined in selected brain regions, including in the anterior pituitary (AP), hypothalamic arcuate nucleus (ARC), preoptic area (POA), and ventro- and dorsomedial nuclei (VMH/DMH). The results indicate that resistin induced a consistent decrease in LeptRb (except in POA) and an increase in SOCS-3 expression during the LD photoperiod in all selected brain regions. In conclusion, the results demonstrate that the action of resistin appears to be strongly associated with photoperiod-driven changes in the leptin signaling pathway, which may underlie the phenomenon of central leptin resistance.

## 1. Introduction

Sheep, which have strong seasonal breeding features, are characterized by circannual rhythms for the secretion of many hormones, food intake and body weight (BW), which makes them an elegant animal example for determining neuroendocrine adaptations of animals to challenging environmental conditions [1]. One such adaptation is the development of leptin resistance/insensitivity. In endocrinology, the phenomenon in which the concentration of a regulatory hormone is high with a lack of response to its action is termed resistance syndrome [2]. Four mechanisms for leptin resistance have been proposed: (1) a decrease in leptin transport from the bloodstream to the hypothalamus (termed peripheral resistance), (2) desensitization of leptin receptors (termed central resistance), (3) suppression of the long isoform of the leptin receptor (LeptRb)-associated signaling pathways by negative regulators such as a suppressor of cytokine signaling (SOCS)-3 (also termed central resistance), and (4) hypothalamic inflammation, which alters the neuronal pathway involved in energy homeostasis [3]. Sheep represent a unique animal model to explore reversible central leptin insensitivity because the ovine hypothalamus and pituitary are resistant to leptin during predictable seasons [4,5]. Since 2004, studies by our laboratory have focused mainly on central leptin resistance caused by the elevated expression of SOCS-3 during the long-day (LD) season [6,7]. Despite the importance of this phenomenon in other species, including humans, as one of the main factors that play a role in the development of metabolic diseases [8], further in-depth experiments are required to elucidate leptin resistance and understand its mechanisms. If significant changes in seasonal leptin insensitivity occur at the hypothalamic–pituitary level in sheep [5,7], questions arise regarding how this effect is mediated.

In the last two decades, numerous adipose-secreted proteins have been shown to behave as hormones and control glucose metabolism and reproductive processes. Leptin has attracted the attention of most of our group since its discovery by Friedman in 1994 [9,10]. This protein, which is consecutively secreted in response to hyperinsulinemia after a meal, reduces appetite, increases energy output, acts at all hypothalamic–pituitary–gonad levels and regulates reproduction. In contrast, resistin, another adipocyte-secreted hormone, induces insulin resistance and hypothalamic leptin insensitivity and strongly increases circulating leptin concentrations in transgenic mice with moderate adipocyte-specific resistin overexpression [11]. Recently, studies by Biernat et al. [12] and Dall’Aglio et al. [13] showed that resistin is engaged in LH (luteinizing hormone) and PRL (prolactin) secretion and uterine function in sheep.

Leptin and resistin both operate on the level of the central nervous system to regulate metabolic processes, suggesting the activation of a few common signaling pathways. A primary site of leptin activity and localization of LeptRb is the arcuate nucleus (ARC) of the hypothalamus [14], but other hypothalamic nuclei are also under the influence of leptin, as shown by research involving microinjections of leptin or demonstrating the spread of its receptor [15,16]. In contrast, the site of action of resistin in the brain has not been thoroughly investigated. However, potential sites of activity were examined using Fos protein detection as a marker of activated neurons [17,18]. These studies suggest that leptin and resistin can activate the same nuclei in the hypothalamus and the medulla [17].

This paper addresses the important question of whether resistin is able to affect leptin secretion from adipose tissue or leptin signaling at the level of the hypothalamus and adenohypophysis and as a result, contribute to the progression of the leptin resistance phenomenon in sheep. Data from other studies have indicated that resistin mediates central leptin resistance in rodents [11] and activates SOCS-3 [19]. Taking into account our previous studies highlighting the role of SOCS-3 in the creation of leptin’s seasonal and pregnancy-induced central resistance in sheep [6,7], we hypothesized that resistin could be a factor that leads to decreased tissue sensitivity to leptin by affecting SOCS-3 and thus LeptRb expression. To verify this hypothesis, the expression of SOCS-3 and LeptRb was determined in selected brain regions: the anterior pituitary (AP), the hypothalamic ARC, the ventro- and dorsomedial nuclei (VMH/DMH), and the preoptic area (POA). Furthermore, in the in vitro experiments, we examined the direct influence of resistin on the secretion of leptin by adipose tissue explants. In these studies, we have also returned attention to the effect of photoperiod, which plays a predominant role in the physiology of seasonally breeding animals.

## 2. Materials and Methods

### 2.1. Animals and Experimental Procedures

The Second Local Ethics Committee on Animal Testing in Krakow, Poland, approved all procedures conducted on animals during the experiments (Protocol No. 109/2018).

A total of thirty Polish Longwool ewes, a breed that exhibits strong seasonal reproduction, were prepared for experiments (ovarectomized with estrogen replacement) as described in detail by Biernat et al. [12]. The animals were aged from 2 to 3 years, weighing 64 ± 4 kg, and were kept in individual pens under natural environmental conditions (longitude: 19° 57 E, latitude: 50° 04 N). The body condition score of the sheep was calculated (BCS = 3, on a scale of 1 to 5) [20], and the animals were fed twice a day at 07:00 and 16:00 with a diet developed to provide 100% of the National Research Institute of Animal Production recommendations for maintenance [21], with constant access to water.

### 2.2. Experimental Design

In both photoperiodic seasons, LD in May and short day (SD) in November, ewes (*n* = 15/season) were randomly assigned to one of three treatment groups (*n* = 5/group/season) and then placed frequently into carts according to a previous report to avoid stress during the experiment [7]. Recombinant bovine resistin (rbresistin) was purchased from CliniSciences (Nanterre, France). The experimental groups were as follows: (1) Control, injected with saline (*n* = 10); (2) R1, injected with a low dose of rbresistin (1.0 µg/kg of body weight (BW); *n* = 10); and (3) R2, injected with a high dose of rbresistin (10.0 µg/kg BW; *n* = 10). The rbresistin doses were chosen based on our previous study [12]. In the morning on the day of each experiment, five randomly chosen sheep were fitted with jugular vein catheters (Central and Peripheral Venous Catheters, Careflow^TM^, Argon, Billmed Sp. z o.o., Warsaw, Poland). At the beginning of the experiment (time 0), saline/rbresistin was injected through the catheter, and 1 h later, blood was collected through the same catheter. Blood samples (5 mL) were dispensed into test tubes containing 150 µL of a solution containing heparin (10,000 IU/mL) and 5% (w/v) EDTA (Ethylene Diamine Tetraacetic) and placed on ice immediately. Plasma was separated by centrifugation and stored at −20 °C until estradiol, resistin, and leptin analyses.

In each photoperiodic season, 1 h after saline/rbresistin infusion and blood sample collection, the animals were humanely euthanized by captive bolt stunning. From the Control group of ewes, perirenal adipose tissue was collected for in vitro experiments. Brains with the infundibulum remaining intact were rapidly removed from the skulls of all ewes and frozen on dry ice. Samples of the AP, the hypothalamic ARC, the VMH/DMH, and the POA, were aseptically isolated from the ewes 10–15 min postmortem. The selected brain regions were collected by removing a tissue block encompassing the hypothalamic–infundibular complex, followed by transection into two halves. An anterior coronal cut was made ~3–5 mm rostral to the optic chiasm, and a posterior coronal cut was made, which contained approximately one-third of the mamillary body. A longitudinal cut parallel to the ventral surface of the brain ~2–3 cm dorsal to the anterior commissure followed. At the same time, the pituitary was harvested from the sella turcica. Isolated tissues were frozen immediately on dry ice for storage at −80 °C. The probability of contamination caused by transferring tissue between samples was eliminated using separate sterile instruments to dissect the chosen area. Samples of brain tissue were rinsed in phosphate-buffered saline (PBS; Laboratory of Vaccines, Lublin, Poland), snap-frozen in liquid nitrogen, and then stored at −80 °C until analysis.

#### Adipose Tissue Preparation and Incubation

Perirenal adipose tissue was aseptically collected from 5 ewes from the Control group per season. Fragments of tissue were transported to the laboratory at 4 °C in Eagle’s sterile medium (Biomed, Lublin, Poland) supplemented with antibiotics (100 µL/100 mL Antibiotic/Antimycotic Solution, SIGMA Chemical Co., St. Louis, USA). Each tissue sample was rinsed three times in Eagle’s medium with antibiotics. The adipose tissue was divided into small pieces, weighing approximately 100 mg each. Then, the samples were randomly assigned to 6-well Corning tissue culture plates (Corning Glass Works, New York, USA) with 2.5 mL of Eagle’s medium. Incubation was performed in a 95% humidified air and 5% CO_2_ atmosphere at 37 °C for a total of 4.0 h. After a 60 min equilibration period, explants were treated with 0 (Control), 1, 10, 100, or 1000 ng/mL doses of rbresistin. The doses of rbresistin were chosen based on a paper by Spicer et al. [22]. All of the treatments were cultured in triplicate, and incubation continued for another 3.0 h. At the end of incubation, one milliliter of medium was collected. Media samples were stored at −20 °C until they were radioimmunoassayed (RIA) for leptin.

### 2.3. Hormone Assays

Concentrations of estradiol in plasma were determined using an ELISA kit (Enzyme-linked immunosorbent assay kit) (DRG Instruments GmbH, Marburg, Germany) according to the manufacturer’s instructions. The inter- and intra-assay precision values exhibited CVs (the coefficient of variations) of 3.9% and 2.6%, respectively, and the assay sensitivity was 1.9 pg/mL. Resistin concentrations were determined using commercially available EIA kits (Cloud-Clone Corp., Katy, USA) according to the manufacturer’s instructions. The inter- and intra-assay precision values exhibited CVs of 6.6% and 2.3%, respectively, and the assay sensitivity was 4.2 pg/mL.

A highly specific ovine leptin RIA using a high-affinity anti-ovine leptin rabbit antibody, anti-rabbit-γ-globulin antisera (double-antibody method) and a recombinant ovine leptin standard as described by Delavaud et al. [23] was employed to determine the leptin concentrations in media and plasma. The sensitivity of the assay was 0.3 ng/mL, the intra-assay coefficient of variation was 2.4%, and the interassay coefficient of variation was 10.7%.

### 2.4. Hormone Statistical Analysis

Data analysis was performed by a series of two-way ANOVAs using SigmaPlot^®^ statistical software (version 11.0; Systat Software Inc., Richmond, CA, USA), preceded by Grubb’s test to identify outliers. The statistical models included the main effects of peptide treatment and season. All data sets with failed tests of normality and/or equal variance were transformed as natural logarithms. If the main effects or their interactions were significant, the Holm–Sidak test was used as a post-ANOVA test to compare individual means. A *P* value < 0.05 was considered to indicate statistical significance. The results are given as the mean ± SEM.

### 2.5. Molecular Analysis

The expression of LeptRb and SOCS-3 was measured using the Real-Time PCR method. Tissue homogenization was performed with a rotor-stator homogenizer (Omni TH, Omni International, Inc., Kennesaw, GA, USA) and single-use tips (Soft Tissue Omni Tip Plastic Homogenizing Probes, Omni International, Inc.). Total RNA was isolated using TRIzol reagent (Ambion Inc., Austin, TX, USA) following the manufacturer’s protocol. Incubation of samples at 42 °C for 2 min with gDNA Wipeout Buffer (QuantiTect Reverse Transcription Kit; Qiagen, Hilden, Germany) was used to eliminate contamination of genomic DNA. Subsequently, to obtain samples of cDNAs by reverse transcription, isolates of RNA (1 µg) were incubated with Quantiscript reverse transcriptase and RT primer mix (QuantiTect Reverse Transcription Kit; as above) at 42 °C for 15 min. The reaction was terminated by heating the samples to 94 °C for 3 min. Amplification of each cDNA was performed in triplicate using an Applied Biosystems 7300 Real-Time PCR System, TaqMan Gene Expression Master Mix, 900 nM concentrations of specific primers corresponding to the target/reference genes (Sequence Detection Primers) and 250 nM concentrations of specific probes corresponding to the target/reference genes (TaqMan MGB Probes) supplied by Life Technologies (Foster City, CA, USA). Primers and probes were designed using Primer Express software v. 2.0 (Applied Biosystems; Foster City, CA, USA) and are characterized in Table 1. The thermal profile of the real-time PCR was as follows: (1) 50 °C for 2 min—initial incubation, (2) 95 °C for 10 min—activation of polymerase, and (3) 40 cycles with denaturation (95 °C for 15 sec) and annealing/elongation (60 °C for 60 sec). The collected data were recorded with the Applied Biosystem 7300 Real-Time PCR System SDS software.

### 2.6. Molecular Data Analysis

The expression levels were calculated using relative quantification (RQ) analysis, and the results were expressed as a function of the threshold cycle (Ct), which is a value corresponding to the fractional PCR cycle number at which the fluorescent signal reached the detection threshold. Data were analyzed using the 2^−ΔΔCt^ method, and Ct values were converted to fold-change RQ values. The RQ values from each gene were used to compare target gene expression across all groups. The mean mRNA expression levels for target genes in each sample were standardized against the expression of a reference gene (cyclophilin; CPH) and expressed relative to the calibrator sample. The variation in the Ct values for CPH among the treatment groups was not significant (*P* > 0.05). The mean ΔCt value for tissue collected from the control group was used as a calibrator to compare the changes in target gene expression among all treatment groups in the indicated season.

Differences in the means were compared with SigmaPlot statistical software (version 11.0; Systat Software Inc., Richmond, CA, USA), using all pairwise multiple comparison procedures (Tukey test), preceded by the determination of a significant F-value. Differences were considered statistically significant when *P* < 0.05.

## 3. Results

### 3.1. Hormone Concentrations

#### 3.1.1. Estradiol Concentrations

The mean circulating concentration of estradiol (mean ± SEM) was 3.6 ± 0.3 pg/mL.

#### 3.1.2. Resistin Concentrations

The mean concentration of circulating resistin (mean ± SEM) in the control group of sheep was 7.8 ± 0.1 ng/mL. The injection of 1.0 µg/kg BW of rbresistin increased (*P* < 0.05) the concentration to 9.5 ± 0.2 ng/mL, and the 10.0 µg/kg BW dose of rbresistin elevated the concentration of resistin to 15.2 ± 0.3 ng/mL (*P* < 0.01). No difference was observed in circulating resistin concentrations between the LD and SD seasons (*P* = 0.08).

#### 3.1.3. Leptin Concentrations in Plasma and Media

Within the Control and all treatment groups, the mean leptin concentration (mean ± SEM) was significantly higher (*P* < 0.001) during the LD season than during the SD season (Figure 1). During the LD season, a lower dose of rbresistin increased the mean leptin concentration compared to the concentrations observed in the Control group (*P* < 0.001) and R2 group (*P* < 0.001). A higher dose of rbresistin increased the circulating leptin concentration compared to that in the Control group (*P* < 0.001); however, the concentration of leptin in that group after resistin treatment was lower than that in the group treated with a lower dose (*P* < 0.01). During the SD season, low and high doses of resistin increased the concentration of leptin (*P* < 0.001) compared to the Control group (Figure 1).

The mean overall concentration of leptin (mean ± SEM) in perirenal adipose tissue explant media was 2.4 ± 0.1 ng/mL and was not affected by season (*P* = 0.9) or culture replicate. Moreover, rbresistin had no stimulatory effect (*P* = 0.2) on leptin secretion during the SD season (Figure 2). During the LD season, however, rbresistin at a dose of 1000 ng/mL stimulated a 3.0-fold increase (*P* < 0.001) in leptin secretion compared to the Control and the other rbresistin-treated explant groups cultured from perirenal adipose tissues (Figure 2).

#### 3.1.4. Leptin Receptor (LeptRb) Expression

In the pituitary leptin receptor, the transcript level increased 2.1-fold (*P* < 0.001) in the R1-group and 1.8-fold (*P* ≥ 0.05) in the R2-group in the LD season. No significant changes (*P* ≥ 0.05) in pituitary LeptRb expression were found between the groups during the SD season (Figure 3). LeptRb transcripts were detected at varying levels in all examined hypothalamic tissues: the ARC, POA, and VMH/DMH during both the LD and SD seasons (Figure 3). During the LD season, within the ARC, LeptRb mRNA levels decreased after rbresistin injection at a high dose compared to the levels noted in the Control (*P* < 0.05) and R1 (*P* < 0.001) groups. No differences (*P ≥* 0.05) were observed during the SD season. Only during the SD season, LeptRb transcripts in the POA increased 8-fold in the R1 (*P* < 0.001) group and 4-fold in the R2 (*P* < 0.05) group compared to the Control group; however, the expression level of LeptRb was 2.2-fold lower in the R2 group (*P* < 0.001) than in the R1-group. The expression of LeptRb in the VMH/DMH decreased in both the R1 (*P* < 0.001) and R2 groups (*P* < 0.001) during the SD season and in response to R2 treatment (*P* < 0.001) during the LD season.

#### 3.1.5. SOCS-3 Expression

In the AP, SOCS-3 transcript levels in the R2 group increased 8.5-fold (*P* < 0.001) compared to those in the Control group and 4.0-fold (*P* < 0.001) compared to those in the R1 group in the LD season. In the SD photoperiod, treatment with a high dose of rbresistin increased SOCS-3 expression 5.8-fold (*P* < 0.001) compared to the control group and 4.8-fold (*P* < 0.001) compared to the R1-group in the pituitary (Figure 4).

The expression of SOCS-3 was detectable in almost all examined hypothalamic tissues, with one exception: no SOCS-3 transcript was detected in the POA during the SD season. During the LD season, within the ARC, SOCS-3 mRNA levels increased significantly (*P* < 0.001) after the injection of a high dose of rbresistin compared to the levels noted after the control treatment (Figure 4). No differences (*P* ≥ 0.05) were observed during the SD season. In the LD and SD seasons, no differences (*P* ≥ 0.05) were found in SOCS-3 transcript levels in the VMH/DMH between rbresistin and control treatments or between the R1 and R2 groups (Figure 4).

## 4. Discussion

The results of the experiments demonstrated for the first time the in vitro and in vivo effects of exogenous resistin on leptin concentrations in sheep. Furthermore, for the first time, resistin dose- and photoperiod-related changes in the expression of the essential components of the leptin signaling pathway have been determined; LeptRb and the key negative regulator SOCS-3 in the anterior part of the pituitary, VMH/DMH, ARC, and hypothalamic nuclei play pivotal roles in appetite regulation, and in the POA, a part of the anterior hypothalamus that possesses leptin neurons and controls energy homeostasis, which is critical to maintaining proper BW and temperature [24].

The first specific aim of our study was to confirm whether central leptin resistance occurs as a consequence of the hyperleptinemia caused by resistin treatment in sheep. Studies carried out by Asterholm et al. [11] demonstrated elevated resistin concentrations and significantly higher circulating leptin concentrations in transgenic mice, suggesting a new mechanism in which metabolic effects are dependent on resistin. In our study, regardless of the treatment (saline or resistin), the circulating concentration of leptin was higher in the period of lengthening days than during the period of shortening days—a phenomenon that is termed physiological central leptin resistance [1]. In seasonal breeding, leptin concentrations are higher in sheep when the days become longer than when days become shorter [4]. This phenomenon has been known since 2001; during the LD period in sheep, the leptin concentration in the bloodstream is elevated by 180% in contrast to the SD [4], but this change is not related to the appetite-lowering effect of leptin. At this time, when readily accessible food is abundant, sheep have an increased appetite and seem to be insensitive to high leptin levels that result from larger amounts of adipose tissue [4]. In our experiments, intravenous treatments with high and low doses of rbresistin significantly elevated leptin concentrations compared to the control treatment during both seasons. These results support the findings of Asterholm et al. [11] in a study of transgenic mice. However, Maillard et al. [25] and Asterholm et al. [11] emphasized the importance of the amount of resistin administered to the animals. The authors noticed a clear difference between the chronic high resistin concentrations in the transgenic mice and those resulting from acute treatment using a pharmacological dose of resistin in another study [11,25]. In the present study, we treated sheep with acute doses of 1.0 or 10 µg of rbresistin per kg of BW. The doses of rbresistin used increased the circulating resistin concentration, though not above 16 ng/mL, which does not exceed the plasma concentration described for cattle and sheep (10–30 ng/mL) [26]. Therefore, we noted that the effect of resistin can be considered physiological. In the present study, we highlight that hyperleptinemia induced by resistin was shown to be a potential hallmark of central leptin insensitivity.

In our ex vivo study, only one dose of recombinant resistin, the highest dose, significantly increased the leptin concentration in the media when cultures were performed during the LD photoperiod, and no other effect was observed. We carried out ovine perirenal adipose explants in vitro using an air–media interface, as we have successfully performed previously [5]. The response of adipose tissue to resistin was expected, particularly during the LD photoperiod; however, only one dose affected the leptin concentration. Many authors have observed that the acute pharmacological effects of hormones often trigger somewhat contradictory results under some circumstances [11,25]. The differences in the action of exogenous resistin between our in vitro and in vivo experiments can be caused by the ligand doses and the conditions of the study. Considering the results of in vitro studies with resistin by other groups, the use of tissue- and species-specific resistin for cell/tissue treatment needs more attention and consideration [25]. In a study on primary culture cells collected from bovine ovaries, Spicer et al. [22] observed a weak or no effect of resistin (recombinant mouse resistin) on thecal cell number and steroid secretion by large and small follicles. Resistin (rat resistin) demonstrated no effects on gonadotropin- or IGF-1-induced steroid secretion from porcine follicles in a 24-h culture compared to control conditions, as noted by Rak et al. [27]. The results of our previous experiments [12] and present experiments indicate that rbresistin is highly bioactive in sheep. The amino acid sequence of resistin is quite homologous in ruminant species; for example, bovine resistin is 91.21% homologous to ovine resistin [12]. There are only eight amino acid differences between them. Furthermore, the effects of resistin depend on binding to its cognate receptors. Although discovered in 2001, the functional receptor for resistin remains elusive, and further research is required to discover receptors for resistin in different species and their specific localization.

Leptin is well known to act mainly in brain regions where a significant number of LeptRb-expressing neurons are present, and areas with exceptionally high LeptRb expression are found in several places in the brain (midbrain, hypothalamus, and brainstem) [15,28]. Every group of LeptRb neurons within the ARC/VMH/DMH circuit supposedly gives certain information relevant to a specific set of inputs and outputs. Each of these brain regions plays a different role in energy homeostasis. The ARC neurons receive information about every small change in insulin, ghrelin and circulating nutrients, as well as leptin, to signal acute peripheral nutritional status. The DMH nuclei do not obtain information about acute changes in the peripheral nutrition status because of the existence of the blood–brain barrier [29] but collect information about the temperature of the body (with POA, where we can find another set of LeptRb neurons), along with information from leptin about long-term energy status [30]. The VMH, which is rich in glucose-sensing neurons, and the LeptRb neurons in this nucleus may also participate in glucose homeostasis regulation [31,32]. The results showed that in those hypothalamic nuclei (with the exception of the POA), exogenous resistin treatment decreased the expression of leptin receptors in a dose-related manner. We reported the approximate effect of leptin on LH secretion in cattle in 2003 [9], with a low dose having a stronger effect on LH compared to a higher dose of peptide; no effect was observed for the highest dose. It seems possible that a high rbresistin concentration can have detrimental effects on the number of LeptRb, and lower expression of the receptor was observed. Furthermore, the differences in resistin action were noticed according to season. Decreasing LeptRb expression with resistin can be speculated to reduce hypothalamic leptin responsiveness and cause central leptin resistance. In our study, acute resistin treatment upregulated LeptRb expression in the POA during the SD season. The neurons in the POA receive a signal about the outside atmospheric conditions and are activated by the low temperature [24]. The POA projects to other nuclei of the hypothalamus, such as the VMH/DMH, which are also engaged in the stimulation of thermogenesis in adipocytes of white and/or brown adipose tissue. POA LeptRb signaling has been indicated to modulate energy output depending on the internal energy state and thus help to maintain homeostasis of BW [24]. In sheep, the elevated expression of LeptRb in the POA during the SD season seems to be required for the regulation of food intake, which depends on the ambient temperature, as demonstrated previously [33,34].

In seasonally breeding animals, the photoperiod primes the sensitivity of the hypothalamic nuclei to the concentration of leptin by modifying the expression of SOCS-3 within the neurons [1]. Another objective of the present study was to evaluate whether exogenous resistin is able to modulate the expression of SOCS-3 in different brain regions in relation to the dose of resistin and day length. Observations by Asterholm et al. [11] indicated how chronic resistin overexpression in transgenic mice affected central leptin resistance and reduced the expression of the inhibitors of leptin signaling—SOCS-3—in the hypothalamic nuclei. Studies carried out previously by our group in sheep have implicated SOCS-3 as a main factor by which leptin negatively affects its own signaling cascade and as a creator of leptin LD central resistance [5,7]. In 2005, Steppan et al. [19] showed, in cultured adipocytes in vitro as well as in adipose tissue, that resistin increases SOCS-3 mRNA expression in a time- and dose-related manner. These results suggest that SOCS-3 affects the capacity of resistin to alienate insulin and leptin activity in cells of adipose tissue [19]. Here, we discuss further evidence that supports the role of a specific member of the SOCS family of proteins, SOCS-3, and its role in resistin-mediated central leptin insensitivity. The expression of SOCS-3 was significantly detected in selected hypothalamic nuclei excluding the POA during the SD season. The lack of detection of SOCS-3 expression in POA during the SD season may result from very weak expression that is commonly demonstrated outside the medio-basal hypothalamus (ARC/VMD/DMH), as shown in field voles, a seasonal species [35], and rodents [36].

During the LD season, SOCS-3 transcript expression was significantly upregulated in the AP, ARC, and POA by a high dose of resistin. A low rbresistin dose caused too much variation in the response of ARC and POA; thus, even though an increase in SOCS-3 expression was noted, the differences between treatment groups were not significant. Based on these findings, we can report that an interaction with resistin contributes to the selective leptin resistance noted at the level of the pituitary gland and hypothalamus in sheep. The results indicate that resistin induced a consistent decrease in LeptRb (except for POA) and an increase in SOCS-3 expression during the LD season in all hypothalamic nuclei. In the AP, resistin increased SOCS-3 levels during both the LD and SD photoperiods, and the LeptRb transcript was increased only during the LD photoperiod. These results suggest that the leptin and resistin signaling pathways are integrated with the LeptRb and SOCS-3 proteins. In agreement with these results, we observed a merger between leptin and resistin-induced expression of SOCS-3 in the pituitary gland and ARC during SDs and in the ARC during LDs in a previous study [7,19]. The SOCS-3 protein was demonstrated to potentially be involved in leptin and resistin interactions. Furthermore, this study indicates how photoperiod affects the effects of resistin on adjusting energy expenditures and energy resources in seasonal organisms.

## 5. Conclusions

To conclude, our data support the hypothesis that resistin elevates leptin concentrations and cooperates with leptin in increasing tissue SOCS-3 expression and lowering LeptRb transcript levels, contributing to decreasing central leptin responsiveness in the brain according to photoperiod. Collectively, these findings show that resistin may be another factor that is involved in central leptin resistance. Further understanding of the mechanisms regulating energy homeostasis and endocrine interactions between leptin and resistin in a large animal model is essential. To date, only a few studies have provided results from in vivo experiments using resistin.

## Figures and Tables

**Figure 1 nutrients-11-02180-f001:**
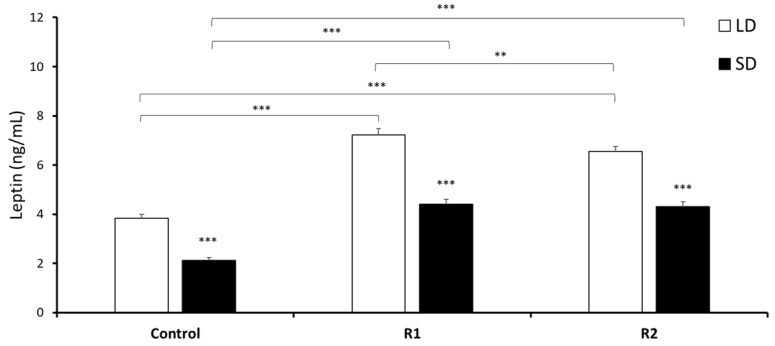
Plasma leptin concentrations. Mean circulating concentrations (±SEM) of leptin in saline- and recombinant bovine resistin-treated groups (R1—low dose and R2—high dose) during the long-day (LD) and short-day (SD) photoperiods. Differences relative to the control or between the other groups are indicated with ** *P* < 0.01 or *** *P* < 0.001.

**Figure 2 nutrients-11-02180-f002:**
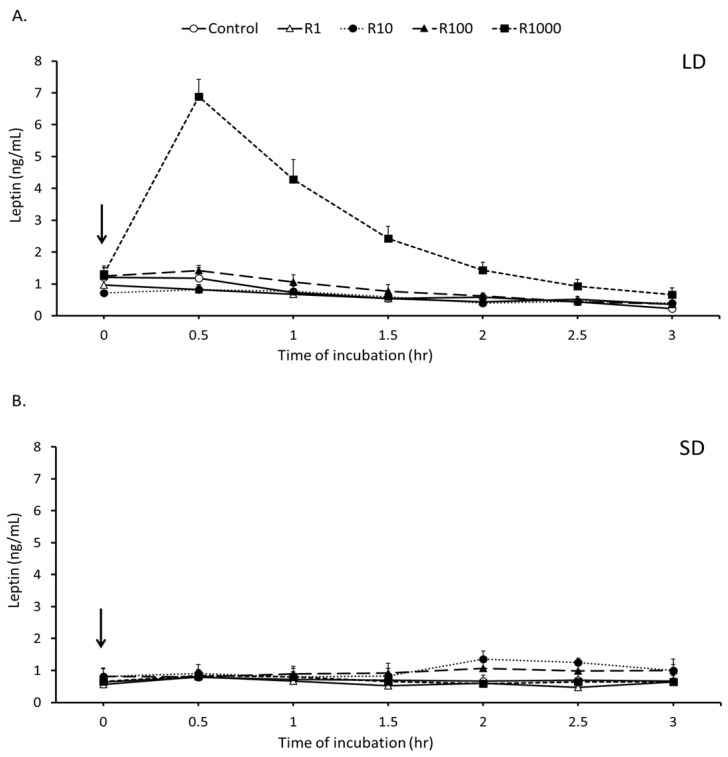
Medium leptin concentrations. Mean concentrations (±SEM) of leptin in media collected during a 3.0 h incubation of ovine perirenal adipose tissue explants obtained during long-day (LD) (panel A) and short-day (SD) (panel B) photoperiods. Adipose tissue explants were incubated with medium containing bovine recombinant resistin (1, 10, 100, or 1000 ng/mL) or medium without hormonal supplementation (Control). Arrows denote 0 (control) h or rbresistin treatment.

**Figure 3 nutrients-11-02180-f003:**
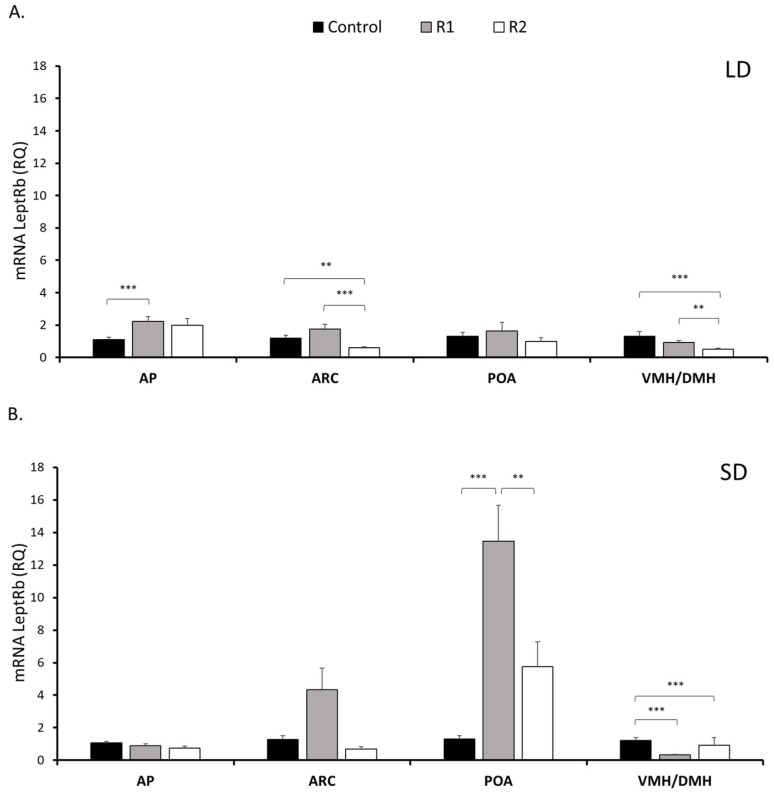
Leptin receptor expression. The mean mRNA expression (±SEM) of the long form of the leptin receptor (LeptRb) in the ovine anterior pituitary gland (AP), arcuate nucleus (ARC), preoptic area (POA) and ventro- and dorsomedial nuclei (VMH/DMH) collected during long-day (LD) (panel (**A**)) and short-day (SD) (panel (**B**)) photoperiods. The expression of LeptRb mRNA is reported in arbitrary units (RQ) relative to cyclophilin mRNA expression and expressed relative to the calibrator sample. Differences relative to the control or between the other groups are indicated with ** *P* < 0.01 or *** *P* < 0.001.

**Figure 4 nutrients-11-02180-f004:**
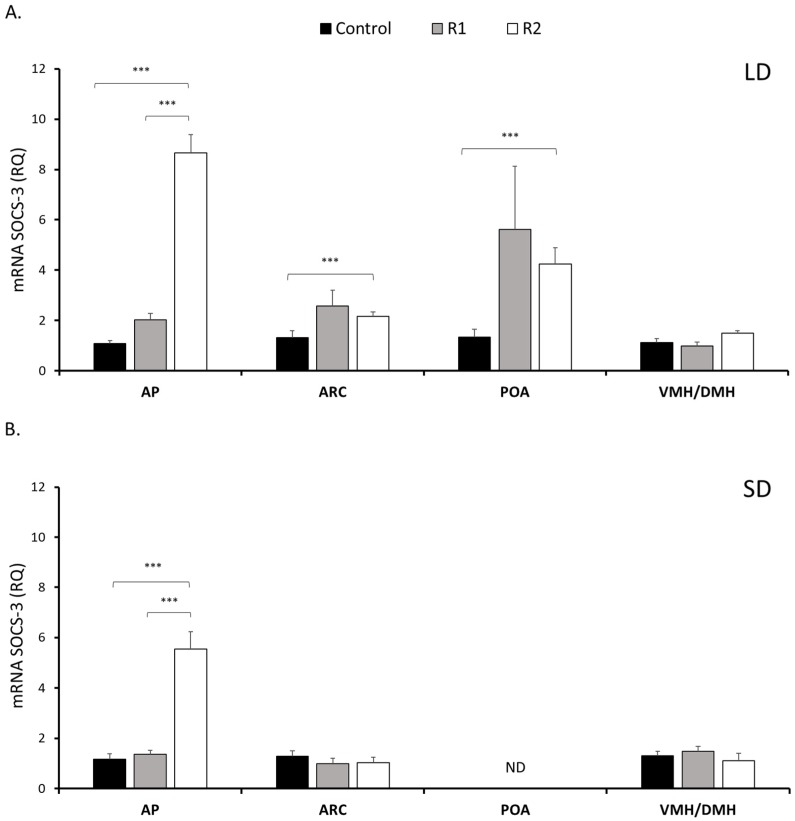
Expression of SOCS-3. The mean expression (±SEM) of a suppressor of cytokine signaling-3 (SOCS-3) mRNA in the ovine anterior pituitary (AP), arcuate nucleus (ARC), preoptic area (POA) and ventro- and dorsomedial nuclei (VMH/DMH) collected during long-day (LD) (panel A) and short-day (SD) (panel B) photoperiods. The expression of SOCS-3 mRNA is reported in arbitrary units (RQ) relative to cyclophilin mRNA and expressed relative to the calibrator sample. Differences relative to the control or between the other groups are indicated with *** *P* < 0.001. Samples in which the expression of the target gene was undetectable are designated with ND.

**Table 1 nutrients-11-02180-t001:** Characteristics of primers and probes used to determine cyclophilin (CPH; reference gene), the long form of the leptin receptor (LeptRb; target gene), and a suppressor of cytokine signaling-3 (SOCS-3; target gene) mRNA expression in sheep.

Gene	Primer Sequence (5′–3′)	Probe Sequence (5′–3′)	Amplicon Size	GenBank Accession Number
**CPH**	CGGCTCCCAGTTCTTCATCA	FAM-CGTTCCGACTCCGC-MGB	64 bp	D14074
	ACTACGTGCTTCCCATCCAAA			
**LeptRb**	CGACGAGGGTGGCATATTTAA	FAM-CAGGAGACAGCCCTC-MGB	63 bp	U62124.1
	CAGACATAACCTGTGAGGATGGAA			
**SOCS-3**	CCTCAAGACCTTCAGCTCCAA	FAM-AGCGAGTACCAGCTGG-MGB	68 bp	NM_174466
	CTTGCGCACTGCGTTCAC			

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
