# Peer review of "Hypothalamic–Pituitary and Adipose Tissue Responses to the Effect of Resistin in Sheep: The Integration of Leptin and Resistin Signaling Involving a Suppressor of Cytokine Signaling 3 and the Long Form of the Leptin Receptor"

_nutrients, 2019, doi:10.3390/nu11092180_

Round 1

Reviewer 1 Report

In this paper, the authors showed that intravenous injection of resistin in sheep increases plasma leptin level, suppresses leptin receptor mRNA in several hypothalamic nuclei, and increases SOCS3 mRNA in those nuclei. These phenomena are interesting, and worth published if resistin-induced leptin resistance has physiological meaning.

Major points:

The final effect of leptin resistance would be increase in food intake, which is not assessed in this study. If venous injection of resistin had been shown to enhance food intake in sheep, that would be much more interesting and important. Why did they measure only markers such as leptin, LepR and SOCS3 but not food intake?

There was no significant difference in serum resistin levels between LD and SD, while leptin level is higher in LD than in SD, which would mean seasonal change in leptin level in sheep is not due to resistin. Why can they conclude "the results demonstrate that the action of resistin appears to be strongly associated with photoperiod driven changes in the leptin signaling pathway"?

Specific points:

Fig. 1 and its legend are difficult to understand. What do "A", "B", "X", "Yy" and "Yz" mean?

While lines 212-214 describes "a higher dose of rbresistin decreased the circulating leptin concentration compared to the Control group ...", the white bar labeled "Yz" in R2 group is higher than the white bar labeled "X" in Control group in Fig. 1, which looks contradictory. Similarly, while lines 214-215 describes "During SD, in the R2 group, lower concentrations of leptin were noted compared to the Control and R1 groups", the black bar labeled "B" in R2 group is higher than the black bar labeled "A" in Control group, which looks contradictory too.

In Fig. 3, LepR looks increased with low-dose resistin in ARC/POA but decreased with high-dose resistin. Is this "biphasic" change fully stated in the result and discussed in the discussion? They describe this as "dose-dependently decreased", which does not look correct.

In SD graph in Fig. 3, comparison among three bars in ARC is shown as not significant, but comparison among three bars in VMH/DMH is significant (P<0.001). Is this correct? At least the difference in ARC bars look quite large.

In Figs. 3 and 4, comparison of LepR/SOCS3 between LD and SD are not performed. Do LepR and SOCS3 change by photoperiod?

Why sheep were ovarectomized? Please explain.

Author Response

We took into account all comments when revising the manuscript.

A detailed list of small changes is presented below, and we indicated changes directly in the text using red font.

Major points:

The final effect of leptin resistance would be increase in food intake, which is not assessed in this study. If venous injection of resistin had been shown to enhance food intake in sheep, that would be much more interesting and important. Why did they measure only markers such as leptin, LepR and SOCS3 but not food intake?

We agree with reviewer that the role of resistin in the regulation of food intake is interesting and important to study. A high leptin concentration due to the existence of leptin resistance is connected to an increase in adiposity and food intake. However, a main aim of our present study was to determine the effect of the photoperiod. Studies were carried out in natural lighting and temperature conditions (line: 94). To determine the effect of resistin on food intake, sheep must be kept in artificial light:dark conditions, and that was not the focus of the present experiments. The next study from our group will be published soon and concentrates on the effects of resistin and the nutritional status of the sheep on leptin resistance, including determination of food intake and many biochemical biomarkers. Previous studies in rats (Cifani et al., 2009) and cattle (Mellouk et al., 2017) indicated that resistin increases food intake, probably through an NPY‐ergic mechanism.

Cifani C, Durocher Y, Pathak A, Penicaud L, Smih F, Massi M, Rouet P, Polidori C. Possible common central pathway for resistin and insulin in regulating food intake. Acta Physiol (Oxf). 2009;196(4):395-400. doi: 10.1111/j.1748-1716.2008.01949.x.

Mellouk N, Rame C, Touzé JL, Briant E, Ma L, Guillaume D, Lomet D, Caraty A, Ntallaris T, Humblot P, Dupont J. Involvement of plasma adipokines in metabolic and reproductive parameters in Holstein dairy cows fed with diets with differing energy levels. J Dairy Sci. 2017; 100(10):8518-8533. doi: 10.3168/jds.2017-12657.

There was no significant difference in serum resistin levels between LD and SD, while leptin level is higher in LD than in SD, which would mean seasonal change in leptin level in sheep is not due to resistin. Why can they conclude "the results demonstrate that the action of resistin appears to be strongly associated with photoperiod driven changes in the leptin signaling pathway"?

In photoperiod-dependent animals (e.g. sheep and Djungurian hamsters), the leptin concentration differs between long and short days, showing circannual changes (Marie et al., 2001; Zieba et al., 2008; Klingerspor 1996, 2000). Circadian changes in leptin concentration have been reported in humans (Licinio et al., 1998). We demonstrated in many papers that the photoperiod is the main factor that influences physiological changes in seasonal sheep, including adiposity, reproduction, metabolic parameters, and the leptin resistance phenomenon, due to an increase in SOCS-3 expression (Zieba et al., 2008; Szczesna et al., 2011). Furthermore, Biernat et al. (2018) recently demonstrated the effect of resistin on LH secretion depending on the season. As it is clear that resistin affects SOCS-3 mRNA expression, in our opinion, the role of photoperiod cannot be ignored in seasonally breeding animals. The differences in SOCS-3 and LeptRb expression levels after resistin treatment between short and long days, particularly in the arcuate nucleus and adenohypophysis, indicated that the photoperiod is involved in the mechanism.

Klingenspor M, Dickopp A, Heldemaier G,  Klaus S. Short photoperiod reduces leptin  gene expression in white and brown adipose tissue of Djungarian hamsters. FEBS Letters 1996;  399:290–294.

Klingenspor M, Niggemann H & Heldmaier G. Modulation of leptin sensitivity by short photoperiod acclimation in the Djungarian hamster, Phodopus sungorus. Journal of Comparative Physiology B 2000;170:37–43.

Licinio J, Negrao AB, Mantzoros C, KaklamaniV,Wong M, Bongiorno PB, et al. Synchronicity of frequently sampled 24-h concentrations of circulating leptin, luteinizing hormone, and estradiol in healthy women. Proc Natl Acad Sci USA 1998;95:2541–6.

Specific points:

1 and its legend are difficult to understand. What do "A", "B", "X", "Yy" and "Yz" mean?

We agree with the reviewer, and changes have been made to clarify the legend.

While lines 212-214 describes "a higher dose of rbresistin decreased the circulating leptin concentration compared to the Control group ...", the white bar labeled "Yz" in R2 group is higher than the white bar labeled "X" in Control group in Fig. 1, which looks contradictory. Similarly, while lines 214-215 describes "During SD, in the R2 group, lower concentrations of leptin were noted compared to the Control and R1 groups", the black bar labeled "B" in R2 group is higher than the black bar labeled "A" in Control group, which looks contradictory too.

We completely agree with the reviewer and apologize for our mistake. The suggested changes have been made in the Results section. In the Discussion section, we interpreted the data correctly.

The following changes were made in the text:

Within the Control and all treatment groups, the mean leptin concentration (mean ± SEM) was significantly higher (P < 0.001) during the LD season than during the SD season (Figure 1). During the LD season, a lower dose of rbresistin increased the mean leptin concentration compared to the concentrations observed in the Control group (P < 0.001) and R2 group (P < 0.001). however A higher dose of rbresistin increased the circulating leptin concentration compared to that in the Control group (P < 0.001); however, the concentration of leptin in that group after resistin treatment was lower than that in the group treated with a lower dose (P < 0.01). During the SD season, low and high doses of resistin increased the concentration of leptin (P < 0.001) compared to the Control group (Figure 1).

In Fig. 3, LepR looks increased with low-dose resistin in ARC/POA but decreased with high-dose Is this "biphasic" change fully stated in the result and discussed in the discussion? They describe this as "dose-dependently decreased", which does not look correct.

The statement “dose-dependent” can be confusing, however we used that term before to describe the effect of leptin on LH. In 2003, in a paper by Zieba et al. titled “Divergent effects of leptin on luteinizing hormone and insulin secretion are dose dependent” we stated that leptin caused a dose-related increase (P < 0.001) in mean concentrations of circulating LH. Stimulation of LH release by leptin was significant at the lowest (141% of control) and middle (122% of control) doses used, but no increase was observed for the highest dose. Dose-dependent action of resistin does not mean that the rise in mRNA expression of LeptRb must be higher depending on dose: the higher dose is, the higher the expression. To clarify that issue we called the effect of resistin a “dose-related effect”.

Appropriate changes have been made in lines 353 and 378.

We discussed this topic further in the Discussion section:

Line: 352-357:

The results showed that in those hypothalamic nuclei (with the exception of the POA), exogenous resistin treatment decreased the expression of leptin receptors in a dose-related manner. We reported the approximate effect of leptin on LH secretion in cattle in 2003 [33], with a low dose having a stronger effect on LH compared to a higher dose of peptide; no effect was observed for the highest dose. It seems possible that a high rbresistin concentration can have detrimental effects on the number of LeptRb, and lower expression of the receptor was observed.

Zieba, D.A.; Amstalden, M.; Maciel, M.N.; Keisler, D.H.; Raver, N.; Gertler, A.; Williams, G.L. Divergent effects of leptin on luteinizing hormone and insulin secretion are dose dependent. Exp. Biol. Med. (Maywood), 2003, 228, 325-330. DOI: 10.1177/153537020322800312

Line: 384-386:

The lack of detection of SOCS-3 expression in POA during the SD season may result from very weak expression that is commonly demonstrated outside the medio-basal hypothalamus (ARC/ VMD/DMH), as shown in field voles, a seasonal species [36], and rodents [37].

In SD graph in Fig. 3, comparison among three bars in ARC is shown as not significant, but comparison among three bars in VMH/DMH is significant (P<0.001). Is this correct? At least the difference in ARC bars look quite large.

We discuss the issue and add a new sentence to manuscript, line 387-390:

During the LD season, SOCS-3 transcript expression was significantly upregulated in the AP, ARC, and POA by a high dose of resistin. A low rbresistin dose caused too much variation in the response of ARC and POA; thus, even though an increase in SOCS-3 expression was noted, the differences between treatment groups were not significant.

In Figs. 3 and 4, comparison of LepR/SOCS3 between LD and SD are not performed. Do LepR and SOCS3 change by photoperiod?

In our study, we concentrated on resistin effects on leptin concentrations and resistin dose- and photoperiod-related changes in the expression of the essential components of the leptin signaling pathway. Taking into account sheep’s seasonality we performed experiments during long and short days. The responses of selected brain areregions to rbresistin were different in May and November (Figures 3 and 4). As reported in our previous experiments (Zieba et al., 2007, 2008, 2015, Szczesna 2011, 2015, Kirsz et al., 2012, 2017) and others (Klingenspor et al., 1996, 2000, Tups et al. 2006, 2006), the close relationship between metabolic hormones (leptin, orexins, and ghrelin) and both length of day and the photoperiod-related signal melatonin in seasonally breeding species was confirmed. Furthermore, the long-day-associated leptin resistance phenomenon is well-known in sheep, and the role of SOCS-3 in that phenomenon was described previously (Zieba et a., 2008, Szczesna et al., 2011, 2015, Szczesna et Zieba, 2015). If we deny the role of photoperiod in the effects of resistin on leptin concentration and signaling pathway-related elements in photoperiod-dependent species, we must deny our group’s and other researchers’ previous results.

Zieba DA, Klocek B, Williams GL, Romanowicz K, Boliglowa L, Wozniak M. In vitro evidence that leptin suppresses melatonin secretion during long days and stimulates its secretion during short days in seasonal breeding ewes. Domest. Anim. Endocrinol. 2007; 33(3): 358-365.

Zieba DA, Kirsz K, Szczesna M, Molik E, Romanowicz K, Misztal T. Photoperiod influences the effects of ghrelin and serotonin receptor agonist on growth hormone and prolactin secretion in sheep. J. Neurol. Neurophysiol. 2015;  6: 301.

Szczesna M., Kirsz K., Kmiotek M., Zięba D.A. Seasonal fluctuations in the steady-state mRNA levels of  suppressor of cytokine signaling-3 (SOCS-3) in the mammary gland of lactating and non-lactating ewes. Small Rumin. Res. 2015;124: 101-104.

Szczesna M, D.A. Zięba. Phenomenon of leptin resistance in seasonal animals: the failure of leptin action in the brain. Domest. Anim. Endocrinol. 2015; 52:60-70.

Kirsz K., M. Szczesna, E. Molik, T. Misztal, A. K. Wojtowicz, D. A. Zieba. Seasonal changes in the interactions between leptin, ghrelin and orexin in sheep. J. Anim. Sci.  2012: 90: 2524-2531.

Kirsz K., Szczesna M., Molik E., Misztal T., Zieba  D.A. Induction of LH and GH secretion by orexin A and ghrelin is controlled in vivo by leptin and photoperiod in sheep. Ann. Anim. Sci.; 2017; 1:155–168.

Tups A, Barrett P, Ross AW, Morgan PJ, Klingenspor M, Mercer JG. The suppressor of cytokine signalling 3, SOCS3, may be one critical modulator of seasonal body weight changes in the Siberian hamster, Phodopus sungorus. J Neuroendocrinol. 2006;18(2):139-45.

Tups A, Ellis C, Moar KM, Logie TJ, Adam CL, Mercer JG, Klingenspor M. Photoperiodic regulation of leptin sensitivity in the Siberian hamster, Phodopus sungorus, is reflected in arcuate nucleus SOCS-3 (suppressor of cytokine signaling) gene expression. Endocrinology 2004;145(3):1185-93.

Why sheep were ovarectomized? Please explain.

In the majority of research on cycling females, the endocrine model is an ovarectomized female with an estradiol implant. Bilateral ovarectomy is performed to eliminate variation due to differences in the concentrations of sex hormones, and estradiol implants are inserted to provide a stable estradiol concentration. When carrying out studies during the estrous and anestrous seasons in seasonally breeding animals, the animals must have a similar steroid status. Usually, to unify females, the estrus synchronization protocol must be provided as we described in our paper from 2008 (Zieba et al., 2008): “During LD, ewes were anovulatory and expressed no signs of oestrus. During SD, oestrous cycles were synchronized using a 14-day treatment with intravaginal progestogen-impregnated sponges (40 mg fluorogestone acetate, FGA, Chronogest; Intervet International, Boxmeer, The Netherlands). Ewes were also injected with a single dose of 500 IU of pregnant mares serum gonadotrophin i.m. (Serogonadotropin, Biowet, Drwalew, Poland) on the day of sponge removal. Oestrous detection was performed twice daily with an adult ram equipped with an apron. Ewes were presented individually to the male. Oestrus was defined as an acceptance of mounting. Experiments were performed when ewes were in the mid-luteal phase (days 7–10) of the estrous cycle.”

However, the reviewers of that paper complained about the comparable status of the ewes and both suggested performing ovarectomy: “Sexually active and inactive. The LD animals were anovulatory and the SD animals were treated with Chronogest plus PMSG to synchronise the oestrous cycle and studied in the mid-luteal phase (p5). The design means that the animal groups were not comparable with respect to sex hormonal status. This can be achieved by studying OVX+oestradiol implanted animals at two seasons. The use of intact animals complicates the interpretation of the seasonal differences in response to leptin“.

Zieba, D.A.; Szczesna, M.; Klocek-Gorka, B.; Molik, E.; Misztal, T.; Williams, G.L.; Romanowicz, K.; Stepien, E.; Keisler, D.H.; Murawski, M. Seasonal effects of central leptin infusion on secretion of melatonin and prolactin and on SOCS-3 gene expression in ewes. J. Endocrinol. 2008, 198, 147–155, DOI:10.1677/JOE-07-0602.

Reviewer 2 Report

Submitted for review, the manuscript contains very interesting and new information regarding resistin potential action in central leptin resitance. This action would be related to the photoperiod. The authors used the Polish sheep as an experimental animal. To avoid influences of hormonal variations, the animals were ovariectomized. They used low and high doses of resistin to evaluate the effects. Blood concentrations in leptin and resistin sheep were evaluated during the study. The levels of gene expression of SOCS-3 and the long leptin receptor isoform were evaluated in different diencephalon target areas and adenohypophysis involved in leptin action. By in vitro experiments, the authors examined the influence of resistin on the secretion of leptin by adipose tissue explants. The authors conclude that resistin could play a role in central insensitivity / resistance to leptin in the long day photoperiod.

The objectives, and material and methods are clearly stated. The results and discussion are significant for the topic. I have the following concerns and suggestions:

Introduction: Although the work focuses on the central role played by resistin, a descriptive part could also be added regarding the role of this molecule in reproduction, e.g.:

Dall'Aglio C, Scocco P, Maranesi M, Petrucci L, Acuti G, De Felice E, Mercati F. Immunohistochemical identification of resistin in the uterus of ewes subjected to different diets: Preliminary results. Eur J Histochem. 2019 May 3;63(2).

L212-214: Please check, as we can see in Fig. 1, during LD, a higher dose of rbresistin decreased the circulating leptin concentration compared only to the group treated with a lower dose. Please also check the L214-215, probably the sentence doesn’t match with the result in Figure 1.

Fig. 1 legend, L219-220: Please clarify, in this sentence blood samples were performed at 10 minutes’ intervals for 4 hrs, there are not explanation in materials and methods. Please, could you provide in a table, the information about all the concentration in the 10 minutes’ time point interval of the 4 hours?

L282-284 Please check grammar

L303 mic

L326 speciess

Please, provide explanation: SOCS3 mRNA wasn’t detect in POA (SD group).

Author Response

Reviewer # 2

We took into account all comments when revising the manuscript.

A detailed list of small changes is presented below, and we indicated changes made directly in the text using red font.

Comment concerning Material and Methods.

Introduction: Although the work focuses on the central role played by resistin, a descriptive part could also be added regarding the role of this molecule in reproduction, e.g.:

Dall'Aglio C.; Scocco P.; Maranesi M.; Petrucci L.; Acuti G.; De Felice E.;, Mercati F. Immunohistochemical Identification of resistin in the uterus of ewes subjected to different diets: Preliminary results. Eur. J. Histochem. 2019,63, 3020, DOI: 10.4081/ejh.2019.3020.

We added some information concerning the role of resistin in sheep reproduction, line 55-64:

Over the last two decades, numerous adipose-secreted proteins have been shown to behave as hormones and control glucose metabolism and reproductive processes. Leptin has attracted the attention of most of our group since its discovery by Friedman in 1994 [9,10]. This protein, which is consecutively secreted in response to hyperinsulinemia after a meal, reduces appetite, increases energy output, acts at all hypothalamic-pituitary-gonad levels and regulates reproduction. In contrast, resistin, another adipocyte-secreted hormone, induces insulin resistance and hypothalamic leptin insensitivity and strongly increases circulating leptin concentrations in transgenic mice with moderate adipocyte-specific resistin overexpression [11]. Recently, studies by Biernat et al. [12] and Dall'Aglio et al. [13] showed that resistin is engaged in LH and PRL secretion and uterine function in sheep.

L212-214: Please check, as we can see in Fig. 1, during LD, a higher dose of rbresistin decreased the circulating leptin concentration compared only to the group treated with a lower dose. Please also check the L214-215, probably the sentence doesn’t match with the result in Figure 1.

The reviewer is correct, and we apologize for our mistake. The changes have been made in the Results section. In the Discussion section, we interpreted the data correctly.

The following changes were made in the text:

Within the Control and all treatment groups, the mean leptin concentration (mean ± SEM) was significantly higher (P < 0.001) during the LD season than during the SD season (Figure 1). During the LD season, a lower dose of rbresistin increased the mean leptin concentration compared to the concentrations observed in the Control group (P < 0.001) and R2 group (P < 0.001). however A higher dose of rbresistin increased the circulating leptin concentration compared to that in the Control group (P < 0.001); however, the concentration of leptin in that group after resistin treatment was lower than that in the group treated with a lower dose (P < 0.01). During the SD season, low and high doses of resistin increased the concentration of leptin (P < 0.001) compared to the Control group (Figure 1).

1 legend, L219-220: Please clarify, in this sentence blood samples were performed at 10 minutes’ intervals for 4 hrs, there are not explanation in materials and methods. Please, could you provide in a table, the information about all the concentration in the 10 minutes’ time point interval of the 4 hours?

We apologize for our mistake. We copied some parts of figure legend from the previous experiment in which we collected blood samples every 10 minutes for 4 hours to determine LH and FSH pulsatility patterns (Biernat et al., 2018). In this experiment, we collected blood twice at the beginning of study and at the end of study to check resistin, leptin and estradiol concentrations.

L282-284 Please check grammar

This section has be corrected.

L303 mic

This section has been corrected.

L326 speciess

This section has been corrected.

Please, provide explanation: SOCS3 mRNA wasn’t detect in POA (SD group).

The following explanation was added to the manuscript, line 383-386:

The lack of detection of SOCS-3 expression in POA during the SD season may result from very weak expression that is commonly demonstrated outside the medio-basal hypothalamus (ARC/ VMD/DMH), as shown in field voles, a seasonal species [36], and rodents [37].

36.   Król, E.; Tups, A.; Archer, Z.A.; Ross, A.W.; Moar, K.M.; Bell, L.M.; Duncan, J.S.; Mayer, C.; Morgan, P.J.; Mercer, J.G.; Speakman , J.R. Altered expression of SOCS3 in the hypothalamic arcuate nucleus during seasonal body mass changes in the field vole, Microtus agrestis. J. Neuroendocrinol. 2007, 19(2), 83-94. DOI: 10.1111/j.1365-2826.2006.01507.x Matarazzo, V.; Schaller, F.; Nédélec, E.; Benani, A.; Pénicaud, L.; Muscatelli, F.; Moyse, E.; Bauer, S. Inactivation of Socs3 in the hypothalamus enhances the hindbrain response to endogenous satiety signals via oxytocin signaling. J. Neurosci. 2012; 32(48),17097-107. DOI: 10.1523/JNEUROSCI.1669-12.2012.

Round 2

Reviewer 1 Report

The manuscript was significantly improved and worth to be published in Nutrients.